# Spectrum of Autoimmune Diseases—Chronic Lymphocytic Inflammation with Pontine Perivascular Enhancement Responsive to Steroids (CLIPPERS)—Clinical Case

**DOI:** 10.3390/medicina59030549

**Published:** 2023-03-11

**Authors:** Agnieszka Meller, Wioletta Pawlukowska, Karolina Machowska-Sempruch, Masztalewicz Marta

**Affiliations:** Department of Neurology, Pomeranian Medical University, 70-110 Szczecin, Poland

**Keywords:** CLIPPERS, brainstem, neuroinflammation, responsive to steroids

## Abstract

Chronic lymphocytic inflammation with pontine perivascular enhancement responsive to steroids (CLIPPERS) syndrome is a rare inflammatory disease of an undetermined aetiology. The condition is characterised by a range of clinical manifestations generally associated with damage to brainstem structures, the cerebellum, with characteristic magnetic resonance imaging (MRI) findings. The main feature is a good clinical and radiological response to glucocorticosteroid (GCS)-based immunosuppressive treatment. The diagnosis of CLIPPERS is difficult and requires extensive differential diagnosis. A specific biomarker in serum or cerebrospinal fluid (CSF) for this disorder is currently unknown. The pathogenesis of CLIPPERS remains poorly understood and its nosological position has not yet been established. Whether CLIPPERS represents an independent, genuine new disorder or a syndrome in the course of diseases with heterogeneous aetiology and/or their precursor stages remains debatable and incompletely clarified. We present a case report of a patient who was diagnosed with CLIPPERS syndrome on the basis of her clinical and radiological features and by performing an extensive differential diagnosis. The patient has been under neurological follow-up for five years.

## 1. Introduction

Chronic lymphocytic inflammation with pontine perivascular enhancement responsive to steroids is a rare inflammatory disease of an unknown aetiology. There is a number of clinical symptoms specific for the disease related mainly to damage to the structures of the brainstem, the cerebellum, with a characteristic picture of magnetic resonance imaging (MRI) [1,2].

The basic feature of the disease is a good clinical and radiological response to immunosuppressive treatment based on glucocorticoids (GCS) [3]. The discontinuation of treatment with GCS often leads to the exacerbation of the disease; therefore, long-term immunosuppressive treatment seems to be necessary to maintain a permanent improvement [4].

The diagnosis of CLIPPERS is difficult and requires extensive differential diagnosis. A specific bio-marker in serum or cerebrospinal fluid (CSF) responsible for this disorder is currently unknown [5]. The pathogenesis of CLIPPERS remains poorly understood and its nosological position has still not been established. Whether CLIPPERS represents an independent, genuine new disorder or a syndrome of heterogeneous diseases and/or their pre-stages remains debatable and is not fully clarified.

## 2. Case Presentation

The paper presents a case report of a patient who was diagnosed with CLIPPERS syndrome based on the clinical and radiological findings and an extensive differential diagnosis. The patient has been under neurological observation for five years.

The first symptoms of the disease appeared in 2016, at the age of 18, in the form of weakness of the lower limbs, which had been increasing for over a year. As a result, in September 2017, the patient was admitted to the Neurology Clinic in Szczecin. Up until then, the patient had not been chronically ill and had not taken any medications on a permanent basis. Their medical history was unburdened.

On admission to the clinic, a neurological examination revealed features of spastic tetraparesis predominant in the lower limbs, with a bilateral Babinski reflex. The patient neither denied sphincter dysfunction nor reported any sensory disturbances. She performed her cohesion tests efficiently. The study did not reveal any abnormalities.

The results of the magnetic resonance imaging of the brain with contrast was normal. However, the magnetic resonance imaging of cervical-thoracic section visualised a slight increase in the signal in the entire section of the spinal cord.

A lumbar puncture was also performed. The general examination of the cerebrospinal fluid showed lymphocytic pleocytosis (12 lymphocytes, no granulocytes, and a normal protein level). The microscopic examination of the cerebrospinal fluid showed a lymphocytic reaction with the presence of activated lymphocytes and transormic forms along with elevated levels of IgG in cerebrospinal fluid. However, the examination did not reveal oligoclonal bands.

The differential diagnosis included the NMO spectrum (neuromyelitis optica), but it did not find anti-AQP-4 (aquaporin-4 antibodies) or anti-MOG antibodies. Systemic connective tissue diseases were taken into account, but the rheumatology panel of the serum and cerebrospinal fluid tests did not show any abnormalities. In differentiation with paraneoplastic syndrome and sarcoidosis, the computed tomography of the chest and abdominal cavity did not reveal any abnormalities; in addition, the panel of onconeural antibodies were normal. Viral diseases were excluded.

The treatment included five boluses of methylprednisolone intravenously. A clinical observation showed an improvement in the neurological condition, which manifested in the form of a decrease in paresis and an improvement in the gait efficiency. The patient was then discharged with a diagnosis of longitudinal autoimmune myelitis of an unclear aetiology. At discharge, encorton, with the initial total dose of 50 mg/day, and azathioprine, 100 mg/day, were applied for chronic treatment.

The patient remained under control in the outpatient clinic for the following months. At the follow-up visit 3 months after the last hospitalisation, the patient had a virtually complete resolution of the previously reported symptoms.

The control MRI of the head and the cervical-thoracic section performed in January 2020 showed a post-inflammatory thinning of the spinal cord without new, active inflammatory foci. It was decided to gradually discontinue the steroids and azathioprine.

During the next outpatient visit after 6 months (July 2020), the doctor stated that the neurological condition of the patient was comparable to the previous visit. In addition, the patient presented non-motor symptoms, i.e., fatigue and concentration disorders. As a result, amantix was included in the treatment, and follow-up neuroimaging tests were ordered. During the next visit to the neurological outpatient clinic after 6 months (January 2021), the patient complained of increasing walking disability that had lasted for about 5 months. The neurological examination showed features of spastic tetraparesis with a predominance of the lower limbs, a bilaterally positive Babinski reflex, as well as sphincter dysfunction. There were neither deviations in the range of cranial nerves nor sensory disturbances; the patient performed her cohesive tests efficiently. The NMR examination of the head and the cervical-thoracic spine in July 2021 revealed in the pons, the medulla oblongata, as well as in the spinal cord (along the entire length) uncountable small pinpoint and speckled foci, which underwent a post-contrast enhancement. Two further small pinpoint foci were visible under contrast enhancement in the genu/rostrum of the corpus callosum on the right side. The radiological description suggested CLIPPERS syndrome (Figure 1 and Figure 2).

The patient was admitted to the PUM Clinic of Neurology (November 2021) for verification of the diagnosis and implementation of the treatment. The differential diagnosis was repeated. NMO spectrum was excluded, the VEP test was normal, and the anti-AQP4 and anti-MOG antibodies were negative. Systemic connective tissue diseases were again excluded (the rheumatologic antibody panel was normal).

The patient was treated with methylprednisolone in a total dose of 5 g intravenously and was recommended to continue oral treatment with prednisone. The initial daily dose equalled 40 mg in the following weeks, with a gradual dose reduction to 20 mg in the forthcoming weeks, and with the inclusion of azathioprine in the dose of 100 mg.

At the scheduled neurological follow-up after 3 months, the study showed a slight improvement in gait; however, the features of spastic tetraparesis with the predominance of the lower limbs persisted. Steroid therapy at a dose of 30 mg and 100 mg of azathioprine was maintained in the treatment. Neuroimaging control was ordered. The control NMR examination of the head in May 2022 showed a complete regression of the previously described changes in the pons and medulla oblongata. The NMR study of the cervical-thoracic section also showed a clear regression in the number and size of previously described foci in the spinal cord. The continuation of the formerly used oral therapy was recommended.

The patient was diagnosed with CLIPPERS syndrome based on the clinical picture, the results of auxiliary tests, a particularly characteristic MRI image, as well as on a good response to steroid therapy.

## 3. Discussion

The guidelines for therapeutic procedures are not clearly defined, which is mainly due to the insufficient number of patients with CLIPPERS syndrome diagnosed so far [6]. It is believed that this disease entity is highly sensitive to steroid therapy, although, in a large number of cases, the remission of symptoms may be incomplete [7,8]. The basis for our diagnosis was primarily the characteristic radiological picture in the form of inflammatory changes within the structures of the pons and medulla oblongata, as well as a good clinical response manifested by the regression of radiological and clinical changes after the use of steroid treatment [1,9]. An additional factor supporting the diagnosis in the case of this patient was the therapeutic failure resulting from the discontinuation of glucocorticoids and azathioprine therapy after 3 years. This contributed to the recurrence of the disease after about 6 months in the form of radiological changes and a deterioration of gait due to progressive paresis.

Because of the small number of cases available, a uniform therapeutic procedure has not been established so far. It seems that the initial treatment of choice should comprise intravenous therapy with high doses of methylprednisolone (e.g., 1 g of methylprednisolone for 5 days), followed by oral GCS therapy [10,11]. When withdrawn or reduced, GCS usually leads to a recurrence of clinical symptoms and inflmmatory activity on imaging studies. The high risk of disease recurrence during GCS reduction is another important reason for the application of long-term immunosuppressive therapy. It usually consists of oral GCS therapy combined with an immunosuppressant (azathioprine, methotrexate, cyclophosphamide, or rituximab) allowing for the use of lower doses of GCS. There was no benefit from treating the patient neither with plasmapheresis nor intravenous immunoglobulin therapy [8,12]. Due to the next relapse of the disease after a discontinuation of the treatment, our patient received an intravenous treatment with methylprednisolone (a total of 5 g was used). Next, the patient received a reduction dose of GCS to the current dose of 20 mg in combination with azathioprine at a dose of 100 mg. Unfortunately, complete clinical regression has not been achieved. A significantly severe pyramidal syndrome with a predominance in lower limbs examination persists. After 3 months from the application of a second relapse treatment, a clear regression of the changes in the control NMR described in the spinal cord was obtained, however, with few foci of post-contrast enhancement.

## 4. Conclusions

Further research is needed to determine the exact nosological position of the disorder, identify potential biomarkers, as well as establish reliable diagnostic criteria and the optimal form and duration of treatment.

## Figures and Tables

**Figure 1 medicina-59-00549-f001:**
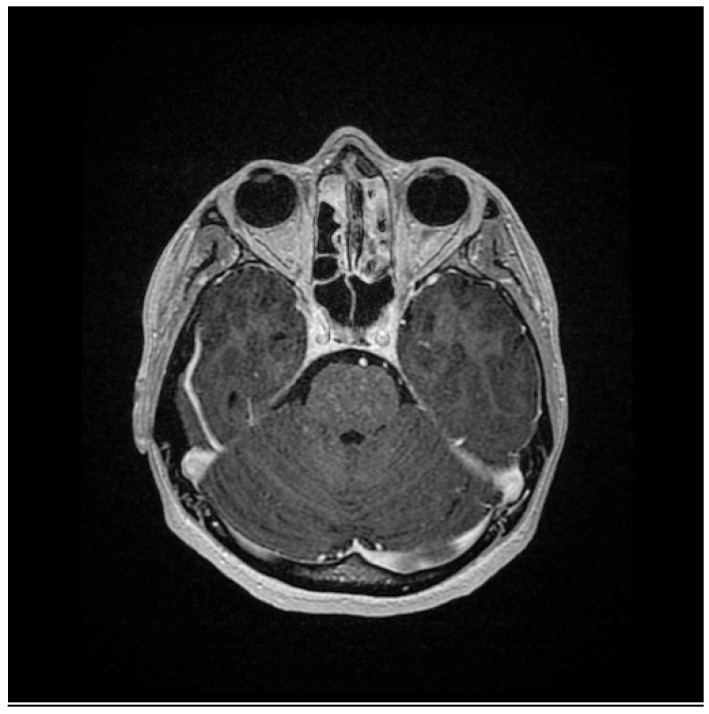
Brain magnetic resonance imaging (MRI) of a patient with CLIPPERS (coronar post-contrast T1-weighted images). MRI shows foci of gadolinium enhancement with a punctate and curvilinear pattern predominantly in the pons.

**Figure 2 medicina-59-00549-f002:**
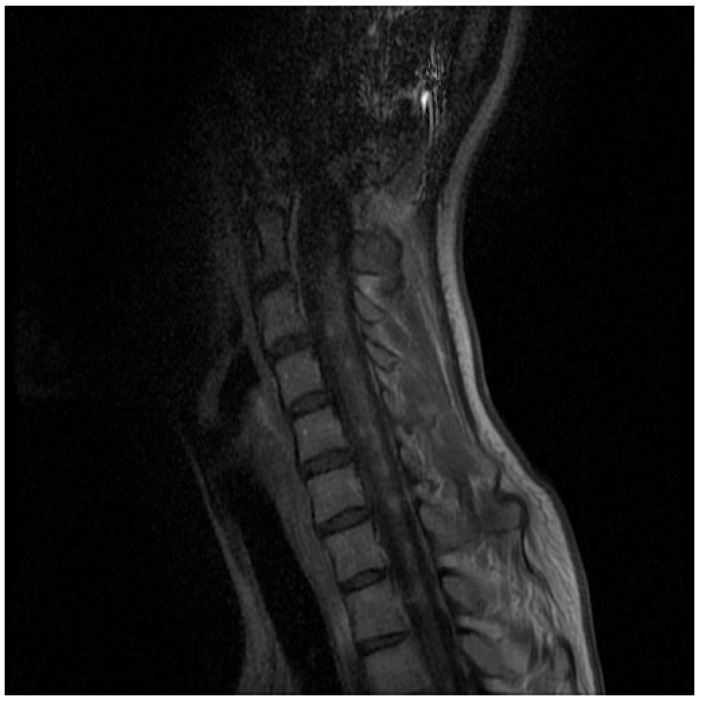
Cervica;-thoracic spine magnetic resonance imaging (MRI) of a patient with CLIPPERS (coronar post-contrast T1-weighted images). MRI shows in the spinal cord (along the entire length) uncountable small pinpoint and speckled foci, which underwent post-contrast enhancement.

## Data Availability

Not applicable.

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
