# Peer review of "Spectrum of Autoimmune Diseases—Chronic Lymphocytic Inflammation with Pontine Perivascular Enhancement Responsive to Steroids (CLIPPERS)—Clinical Case"

_medicina, 2023, doi:10.3390/medicina59030549_

Round 1

Reviewer 1 Report

Dear Authors,

I have a few queries please address it.

1. In the abstract line 12-13, change the brackets words to CLIPPERS.

2. In refence to the statement in line 34-35, The etiology of disease was still unknown, but how the medication of GCS was giving. On what basis these glucocorticoids are giving.

3. Can the authors provide any MRI or CT images of disease progression which was conducted in the last five years of the case study.

4. Please provide images of control MRI of the head and the cervico-thoracic section performed in January 2020 showed post-inflammatory thinning of the spinal cord.

5. Examination of the cerebrospinal fluid showed lymphocytic pleocytosis, provide the evidence in you case study

Reviewer 2 Report

The case report “Spectrum of autoimmune diseases - CLIPPERS - (Chronic lym- 2 phocytic inflammation with pontine pervascular enhancement 3 responsive to steroids). Clinical case” by Meller er al., presents interesting clinical observation with CLIPPERS. Authors have well written and represented clinical observation with patient for the course of five years. I have few minor comments below to further attract clinicians and reader;

1.     If feasible authors should incorporate important MRI images during the time course.

2.     Table or figure depicting the clinical findings throughout the course of patient treatment would be helpful to clearly understand the clinical course of CLIPPERS.

Round 2

Reviewer 1 Report

I congratulates to all the author's to accept the manuscript after the revision.

Please include the images of the MRI scan in the manuscript, so that the readers can understand the disease conditions and improvement. 
